# The Role of 18F PSMA-1007 PET/CT in the Staging and Detection of Recurrence of Prostate Cancer, A Scoping Review

**DOI:** 10.3390/cancers17061049

**Published:** 2025-03-20

**Authors:** David Armany, Lequang Vo, Duncan Self, Sriskanthan Baskaranathan, Tania Hossack, Simon Bariol, David Ende, Henry Hyunshik Woo

**Affiliations:** 1Department of Urology, Blacktown Hospital, Sydney, NSW 2148, Australia; david.armany@health.nsw.gov.au (D.A.); lequang.vo@health.nsw.gov.au (L.V.); duncan.self@health.nsw.gov.au (D.S.); sriskanthan.baskaranathan@health.nsw.gov.au (S.B.); tania.hossack@health.nsw.gov.au (T.H.); simon.bariol@health.nsw.gov.au (S.B.); david.ende@health.nsw.gov.au (D.E.); 2Blacktown-Mount Druitt Clinical School, Western Sydney University, Sydney, NSW 2148, Australia

**Keywords:** PSMA-1007, biochemical recurrence, prostate cancer, definitive therapy, PSMA-PET, initial staging, relapse prostate cancer, metastatic prostate cancer

## Abstract

Prostate cancer is amongst one of the most diagnosed malignancies affecting adult men worldwide and exhibits broad clinical behavior. Early and accurate staging at initial diagnosis and for patients with biochemical recurrence is vital to guiding treatment decisions and improving oncological outcomes. This scoping review analyzes the current evidence landscape surrounding the role of 18F-PSMA-1007 PET/CT in the initial staging and detection of recurrent disease. This review highlights that 18F-PSMA-1007 PET/CT can upstage prostate cancer on initial staging and, when combined with multiparametric MRI, can downstage equivocal PIRADS 3 lesions. With regards to biochemical recurrence, 18F-PSMA-1007 PET/CT demonstrates a reasonable ability to detect recurrence at lower PSA levels compared to other commonly used radiotracers and can influence treatment decisions regarding salvage therapies. Notable limitations include poor specificity for bone lesions and inconsistent urinary excretion patterns. Further prospective multicenter trials are required to clearly delineate its role in Prostate cancer staging, however is likely useful as a second-line imaging modality in which locoregional recurrence is highly suspected with low PSA levels given its superior sensitivity performance.

## 1. Introduction

Prostate cancer (PCa) is amongst one of the most diagnosed malignancies affecting adult men worldwide, resulting in significant health implications and burdens on healthcare systems [1]. The accurate diagnosis, staging and early detection of disease recurrence are vital in guiding early interventions to ensure good disease-related patient outcomes [2]. Prostate cancer exhibits broad clinical behavior ranging from low volume low-risk disease requiring minimal clinical intervention, to aggressive phenotypes with significant metastatic potential [3]. Therefore, the accurate assessment regarding the extent of prostate cancer disease is critical in determining the appropriate management approach and predicting patient outcomes and prognosis. Imaging modalities have played a central role in the assessment of initial tumors, staging and detection of disease recurrence, particularly in the context of biochemical recurrence following definitive therapy. Traditional imaging modalities, including Computed Tomography (CT), Multiparametric Magnetic Resonance Imaging (mpMRI), and Whole-Body Bone Scans (WBBS); have served as the cornerstone for the staging and detection of prostate cancer [4]. However, these are not without limitations, particularly in the detection of early metastatic or recurrent lesions in the context of biochemical recurrence [5]. In this context, there has been a growing need for more sensitive and specific imaging techniques that will permit appropriate management-based decisions.

The advent of molecular imaging, particularly Prostate-Specific-Membrane-Antigen (PSMA) targeted Positron Emission Tomography (PET), has marked a paradigm shift in the imaging landscape of PCa. PSMA is a transmembrane glycoprotein that is overexpressed in prostate cancer cells with limited expression in normal tissue. This overexpression, along with its cell surface localization, makes PSMA an ideal target for emerging molecular imaging technologies [6]. The Fluorine-18 PSMA-1007 (F18 PSMA-1007 PET/CT) radiotracer has emerged as a recent promising agent in both the staging and detection of recurrent disease due to its early promising results suggesting high diagnostic accuracy [7]. The 18F PSMA-1007 is a fluorine-18 labeled PSMA ligand that has gained recent attention secondary to its favorable imaging properties, including high tumor cell uptake, low urinary excretion primarily due to its hepatobiliary clearance, and prolonged lesion retention [8]. These imaging characteristics are advantageous over other PSMA-targeted radiotracers, such as gallium-68 PSMA-11 (68Ga-PSMA-11), particularly for the detection of pelvic and periprostatic lesions where urinary activity can obscure the findings during reporting [9].

Within the context of primary staging, 18F-PSMA-1007 PET/CT is promising in its ability to detect both locoregional and distant metastatic disease [10]. Whilst conventional imaging modalities are widely used with strong supporting evidence, they often lack the ability to detect micro-metastatic disease or lesions in atypical locations [11]. By contrast, 18F-PSMA-1007 PET/CT can identify small lymph node metastases and early bone lesions, not yet apparent on CT or mpMRI. This capability, although yet to be fully explored, has the potential to improve oncological outcomes through greater diagnostic accuracy and precise treatment planning [12]. Moreover, the detection of recurrent disease following definitive therapy represents another critical application for the 18F-PSMA-1007 PET/CT. Biochemical recurrence post definitive treatment is a clinical scenario often encountered by treating urooncologists and poses subsequent management challenges. Whilst PSA kinetics including the PSA doubling time and velocity can provide an indication regarding disease recurrence, imaging is essential in localizing disease recurrence sites and thus guiding available salvage therapies [13,14]. The role of 18F-PSMA-1007 PET/CT is novel in these clinical scenarios; however, its ability to detect prostate cancer in patients with low PSA levels, a common limitation of conventional imaging techniques, may enable earlier and more targeted interventions [5].

The clinical utility of F18 PSMA-1007 PET/CT in the staging and detection of recurrence of PCa is an area of active research with accumulating early evidence supporting its clinical value and establishing its role in the management of prostate cancer disease. This scoping review’s purpose is to summarize and disseminate the available current research evidence about the impact of this PSMA radiotracer in guiding ongoing prostate cancer treatment.

## 2. Materials and Methods

### Search Strategy

A comprehensive literature review was conducted in September 2024, adhering to the Arksey and O’Malley (2005) framework for scoping review methodology. Relevant abstract titles were retrieved from PubMed/MEDLINE, EMBASE, EBSCO, and the Cochrane Central Register of Controlled Trials (CENTRAL) databases using a combination of the following keywords: “PSMA-1007”, “Biochemical Recurrence”, “Prostate Cancer”, “Definitive Therapy”, and “PSMA-PET”. Boolean operators were employed to further refine the search using terms such as “initial staging”, “relapse prostate cancer”, and “metastatic prostate cancer”. This systematic search approach identified a total of 404 publications (Figure 1).

Duplicates and irrelevant articles were excluded (n = 343), leaving 61 abstracts for screening based on a predefined inclusion criterion: (1) studies involving patients with PCa undergoing initial staging or detection of recurrent PCa; (2) publications in English; and (3) studies must utilize F18-PSMA-1007 PET/CT, with or without comparisons to other imaging modalities. Following this screening process, 26 studies were excluded, resulting in 35 articles included in this scoping review.

All study designs, excluding systematic reviews and meta-analyses, were considered. Key findings were extracted and organized into two thematic tables: studies addressing initial staging (Table 1) and those focusing on restaging in the context of PCa recurrence following definitive therapy (Table 2). This approach facilitated a comprehensive mapping of the current evidence on the role of 18F-PSMA-1007 PET/CT in staging and detecting recurrent prostate cancer. The analysis involved comparing findings across the studies within these two main themes and identifying relevant categories to synthesize and contextualize the available knowledge.

## 3. Results

### 3.1. Article Characteristics

Thirty-four articles reported on findings from quantitative studies (97.1%), twenty-one of which were retrospective studies (60%) and thirteen were prospective studies (37.1%). One study was a randomized controlled trial (2.9%). A total of 16/35 studies included in this review pertained to the initial staging of patients with prostate cancer, whilst 19/35 evaluated 18F-PSMA-1007 PET/CTs diagnostic accuracy for patients with recurrent prostate cancer following definitive therapy. Study sample sizes ranged from 10 to 584 participants (mean ± std; 110.2 ± 113.1) and were conducted in China (n = 11), Netherlands (n = 4), India (n = 1), Finland (n = 1), Australia (n = 1), Germany (n = 6), South Africa (n = 2), Greece (n = 1), Switzerland (n = 2), Japan (n = 1), Belgium (n = 1), Poland (n = 2), France (n = 1), and Spain (n = 1). Data collection methods involved medical records database review, and imaging interpretation by professional nuclear medicine specialists. Correlation with PSMA PET imaging findings was often performed with radical prostatectomy histopathological specimens in the case of initial staging and with other imaging modalities, serum PSA, and Gleason scores in the case of recurrent disease.

### 3.2. The Role of 18F-PSMA-1007 PET/CT in the Initial Staging of Prostate Cancer

#### 3.2.1. Accuracy of 18F-PSMA-1007 with Regards to Histological Validation

The accuracy of 18F-PSMA-1007 PET/CT has been validated in two studies using histological confirmation. Luo et al. (2023) performed a retrospective analysis of 117 patients with confirmed prostate cancer undergoing 18F-PSMA-1007 PET/CT prior to radical prostatectomy. Postoperative radical prostatectomy histopathological specimens were used as a reference standard and a detection rate of 96.6% was calculated for intraprostatic lesions detected preoperatively by 18F-PSMA-1007 PET/CT. Sensitivity and specificity were 91.2% and 89.8%, respectively. Additionally, Bai et al. (2022) analyzed 257 patients demonstrating that detection rates of 18F-PSMA-1007 PET/CT were positively correlated with clinical risk factors, including PSA levels, prostate biopsy Gleason scores, and the D’Amico risk classifications.

#### 3.2.2. Comparative Imaging Evaluation Between 18F-PSMA-1007 PET/CT and Other Imaging Modalities

Nine studies conducted comparative analyses evaluating 18F-PSMA-1007 PET/CT against other alternative imaging modalities (Liu et al., 2022; Chandekar et al., 2023; Wondergem et al., 2021; Privé et al., 2021; Annttinen et al., 2021; Ye et al., 2024; Kesch et al., 2017; Chen et al., 2023). Chandekar et al. (2023) prospectively recruited 40 prostate cancer patients comparing 18F-PSMA-1007 PET/CT and 68Ga-PSMA-11 PET/CT. Additional regional lymph nodes and skeletal lesions were detected using 18F-PSMA-1007 PET/CT compared to 68Ga-PSMA-11 PET/CT, potentially upstaging two patients. Privé et al. (2021) retrospectively analyzed 53 patients demonstrating higher diagnostic accuracy of 18F-PSMA-1007 PET/CT for staging seminal vesical invasion (90%) compared to mpMRI (76%). However, mpMRI demonstrated greater accuracy with regards to extracapsular extension (90% vs. 57%). Annttinen et al. (2021) compared 18F-PSMA-1007 PET/CT to conventional imaging (bone scans, CTs and whole-body MRIs) in 79 patients with primary prostate cancer. 18F-PSMA-1007 PET/CT outperformed all other modalities in detecting distant metastatic lesions and influenced clinical decision-making in 18% of cases. Liu et al. (2022) reported higher sensitivity and negative predictive values for 18F-PSMA1007 PET/CT, though mpMRI showed greater specificity for positive lymph node metastasis. Wondergem et al. (2021) performed a matched comparison of 240 prostate cancer patients undergoing either 18F-PSMA-1007 PET/CT (n = 120) or 18F-DCEFPyl PET/CT (n = 120). 18F-DCEPyL PET/CT had a higher inter-reader agreement and fewer equivocal skeletal lesions; however, 18F-PSMA-1007 demonstrated superior diagnostic accuracy for prostatic or prostatic fossa lesions.

Several studies explored the combined use of 18F-PSMA-1007 PET/CT with other imaging modalities. Privé et al. (2024) conducted a prospective study of 75 patients reporting that adding 18F-PSMA-1007 PET/CT to mpMRI improved diagnostic accuracy for equivocal PIRADS 3 lesions when correlated with target biopsy results, thereby ruling out clinically significant prostate cancer in 93% of cases. Additionally, Awenat et al. (2021), in a systematic review of 369 patients, suggested dual imaging with mpMRI may enhance staging accuracy, particularly for patients with negative findings on standard imaging.

#### 3.2.3. Neoplastic Site-Specific Analysis

The ability of 18F-PSMA-1007 PET/CT to detect PCa lesions varies across anatomical locations, including the pelvis (primary tumor, and lymph nodes), distant lymph nodes and bony lesions. Zheng et al. (2023) conducted a retrospective study of 152 patients exploring its ability to predict pathological upgrading from systematic biopsy to radical prostatectomy. 26.97% of cases had Gleason score upgrading following imaging, suggesting that PET/CT can reveal previously undetected or underestimated lesions. Prostate volume and total PSMA uptake as measured by the SUVmax were independent predictors of pathological upgrading, thereby suggesting that PSMA-1007 is effective in identifying primary tumors with more aggressive histological features.

The detection of lymph node metastases within the pelvic region is essential for the accurate staging of prostate cancer. Guo et al. (2023) conducted a retrospective analysis of 107 patients with gray-zone PSA levels (4–10 ug/L) who underwent 18F-PSMA-1007 PET/CT. The study demonstrated higher levels of PSA Density (PSAD) and SUVmax in patients diagnosed with prostate cancer compared to those without malignancy. This distinction enabled 18F-PSMA-1007 PET/CT to differentiate between malignant and benign pelvic lymph nodes, thereby enhancing its sensitivity in detecting micrometastatic disease within pelvic lymph nodes. As such, its diagnostic capability proved valuable in cases where PSA levels alone were inconclusive to establish the presence of metastatic involvement.

Additionally, 18F-PSMA-1007 PET/CT’s capability for detecting bone metastases is contentious. Chen et al. (2023) compared 18F-PSMA-1007 PET/CT with 99Tcm MDP SPECT/CT in a retrospective study analyzing 77 patients, reporting sensitivity rates of 100%, and specificity of 97.14% for the detection of bone metastases. This finding contrasted with Hagens et al. (2022) earlier retrospective study of a larger cohort of 584 patients that found higher interobserver variability for 18F-PSMA-1007 PET/CT, particularly in the assessment of bone metastases when compared to other PSMA-radiotracers. This was later congruent with Huang et al. (2023) study that highlighted higher false-positive rates for bone lesions reported for patients undergoing 18F-PSMA-1007 PET/CT when compared to 68Ga-PSMA-11 PET/CT. Additionally, Ye et al. (2024), in a study involving 41 patients who received both 18F-PSMA-1007 PET/CT and multiparametric Magnetic Resonance Imaging (mpMRI), found superior diagnostic accuracy (95.1%) for 18F-PSMA-1007 PET/CT compared to 82.9% for mpMRI.

Furthermore, 18F-PSMA-1007 was shown to influence clinical treatment decisions, adding to its prognostic value. Li et al. (2020) retrospectively evaluated 18 patients with newly diagnosed prostate cancer who underwent 18F-PSMA-1007 PET CT with 13/18 finding metastatic lesions (n = 72.2%) and subsequent treatment decision changes in 8/17 patients (n = 47.1%), after imaging.

### 3.3. The Role of 18F-PSMA-1007 PET/CT in the Detection of Recurrent Prostate Cancer

#### 3.3.1. PSA-Based Stratification

The diagnostic performance of 18F-PSMA-1007 PET/CT in the detecting biochemical recurrence (BCR) of prostate cancer was evaluated in 8 studies (Zhou et al., 2022; Mingels et al., 2022; Watabe et al., 2021; Ahmadi Bidakhvidi et al., 2021; Giesel et al., 2018; Giesel et al., 2019; Poterszman et al., 2024; Sprute et al., 2021). Three studies employed prospective quantitative designs with study samples between 12 and 251 patients and reported high 18F-PSMA-1007 PET/CT detection rates, particularly with PSA levels > 1.0 ng/mL, with detection rates subsequently declining at lower PSA levels (Gisel et al., 2018,2019 and Watabe et al., 2021). Retrospective studies such as Zhou et al. (2022) and Ahmadi Bidakhvidi et al. (2021), employed medium-sized study samples (71 and 137 patients, respectively), and highlighted biochemical predictors including PSA velocity and Gleason scores for scan positivity. Poterszman et al. (2024) retrospectively analyzed 71 patients and emphasized stratified detection rates, signifying higher positivity rates in high-risk groups with median biochemical recurrent PSA levels of 1.43 ng/mL (IQR 0.736–2.77). Approximately 40% of patients with PSA < 0.5 ng/mL had at least one PSMA-positive lesion. Additionally, Mingels et al. (2022) reported on 177 patients providing a region-specific analysis of 18F-PSMA-1007 PET/CT’s diagnostic accuracy. Overall sensitivity and specificity rates were higher (95% and 89%, respectively), the PPV was 97% for local recurrence and 93% for pelvic lymph nodes but 79% for bone lesions, demonstrating a diagnostic weakness. Additionally, Sprute et al. (2021) looked at 96 patients who underwent 18F-PSMA-1007 PET/CT for both primary staging and biochemical recurrence, demonstrating high specificity for lymph node staging, with lesion-based sensitivity of 81.7% for nodes > 3 mm in size.

The detection of prostate cancer recurrence at lower PSA levels was also studied. Three prospective studies examined 18F-PSMA-1007 detection efficacies of patients with low PSA levels in the setting of biochemical recurrence (Tian et al., 2020; Witkowska-Patena et al., 2020 and Garcia et al., 2024). Tian et al. (2020) demonstrated detection efficacies of 50%, 78.6%, and 88.2% for PSA levels 0.2–0.5 ug/L, 0.5–1.0 ug/L and 1.0–2.0 ug/L, respectively. Witkowska-Patena et al. (2020) found detection rates of 39%, 55% and 100% for PSA < 0.5, 0.5 to <1.0 and 1.0 to <= 2.0 ng/mL, respectively, finding that an increase of 30% in detection rates was correlated with a PSA rise of just 0.1 ng/mL. Garcia et al. (2024) prospectively enrolled 35 patients who underwent simultaneous 18F-PSMA-1007 PET and integrated MRI following prostatectomy with BCR PSA levels < 0.5 ng/mL, demonstrating a 14.3% improvement of detection rates and influencing treatment clinical decisions in 80% of cases. One retrospective study, Lengana et al. (2022), enrolled 46 patients observing detection rates of 31.3%, 33.3%, 55.6% and 72.2% for PSA levels < 0.5, 0.5 - 1, 1–2, >2, respectively.

#### 3.3.2. Comparative Imaging Results for Recurrent Prostate Cancer

Six review articles compared 18F-PSMA-1007 PET/CT to other imaging methods within the context of biochemical recurrence (Lengana et al., 2021; Panagiotidis et al., 2023; Rauscher et al., 2020; Alberts et al., 2022; Hoffmann et al., 2022; Witkowska-Patena et al., 2019). Three studies compared 18F-PSMA-1007 PET/CT with 68Ga-PSMA-11 PET/CT in BCR patients with sample sizes between 21 and 264. (Hoffmann et al., 2022; Lengana et al., 2021; Rauscher et al., 2020). Lengana et al. (2021) demonstrated superior sensitivity and specificity for 18F-PSMA-1007 PET/CT (88.9%, 100%, respectively) compared to 68Ga-PSMA-11 PET/CT (44.4% and 83.3%) in patients with a mean PSA 2.55 ± 3.1 (IQR 0.05–8.93). However, Rauscher et al. (2020) in 102 patients with a median PSA 0.87 ng/mL (IQR 0.20–13.59) revealed similar lesion detection rates between the two tracers (124/369 for 18F-PSMA-1007 PET/CT vs. 126/178 for 68Ga-PSMA-11 PET/CT) but noted that 18F-PSMA-1007 PET/CT had greater uptake in benign lesions, demonstrating the need for sophisticated reader training. These discrepancies likely result from the mean PSA of patients analyzed between the two studies. Hoffman et al. (2022) retrospectively enrolled 136 undergoing 68Ga-PSMA-11 PET/CT and 128 undergoing 18F-PSMA-1007 PET/CT for a matched comparison, highlighting that median and mean PSA levels significantly influenced 18F-PSMA-1007 positivity rates, a trend that was not seen in the 68Ga-PSMA-11 study cohort. Two studies compared 18F-PSMA-1007 PET/CT to 18F-Choline PET/CT (Panagiotidis et al., 2023 and Witkowska-Patena., 2019). In a multicenter randomized trial with 186 participants, Panagiotidis et al. (2023) observed superior imaging detection rates for 18F-PSMA-1007 compared with 18F-Choline PET/CT (84% vs. 69%, respectively) and better performance at lower PSA levels. Witkowska-Patena et al. (2019) prospectively enrolled 40 patients after radical treatment with PSA levels < 2.0 ng/mL. All patients underwent both 18F-PSMA-1007 PET/CT and 18F-Choline PET/CT. 18F-PSMA-1007 had a higher positive lesion rate (60% vs. 5%) and a lower equivocal rate of (12.5% vs. 37.5%) compared to 18F-Choline, upgrading 70% of 18F-Choline scans, thus indicating a greater prognostic value.

#### 3.3.3. Salvage Therapy and the Role of 18F-PSMA-1007 PET/CT

The application of 18F-PSMA-1007 PET/CT in salvage therapy for prostate cancer has also been investigated, with emerging evidence supporting its role in refining treatment strategies through improved lesion detection and clinical decision-making. Garcia et al. (2024) demonstrated that 18F-PSMA-1007 PET/MRI enhanced detection rates by 14.3% compared to either modality alone and influenced treatment decisions in 80% of patients. Notably, these findings lead to radio-guided salvage radiotherapy (RT) in 57% of patients and the initiation of systemic therapy in 20%, predominantly due to the detection of widespread metastatic disease. This underscores the ability of 18F-PSMA-1007 to accurately localize recurrence sites, even at minimal PSA elevations (<0.5 ng/mL in this study), thereby enabling more precise salvage treatment strategies. The enhanced sensitivity of 18F-PSMA-1007 PET/CT in the detection of recurrent prostate cancer at low PSA levels is particularly relevant for early intervention in salvage therapy. Lengana et al. (2022) demonstrated reasonable detection rates at low levels of BCR, suggesting the capability of 18F-PSMA-1007 to detect recurrence at earlier time intervals, allowing for the timely initiation of salvage therapies before disease progression reaches a further advanced stage. Furthermore, Witkosa-Patena et al. (2019) study found that when compared to 18F-Choline PET/CT, 18F-PSMA-1007 PET/CT led to a 70% upgrade rate of previously negative or equivocal scans demonstrating its effectiveness for salvage therapy planning by providing a more definitive localization of recurrence disease, reducing the likelihood of undertreating or overtreating patients. This definitive localization of recurrence is fundamental in optimizing salvage radiotherapy targeting to maximize tumor control whilst minimizing toxicity. Poterszman et al. (2024) demonstrated that PSMA-1007 PET/CT positivity rates for detection of recurrence were significantly higher in high-risk patients compared to low-risk, suggesting that 18F-PSMA-1007 PET/CT can serve as an important prognostic tool in stratifying patients for more aggressive salvage therapy approaches.

Beyond its role in radiotherapy guidance, 18F-PSMA-1007 PET/CT also plays a crucial role in determining the need for systemic salvage therapy. Hoffmann et al. (2022) analyzed 264 patients with PSA relapse post-radical treatment, comparing 68Ga-PSMA-11 PET/CT with 18F-PSMA-1007 PET/CT. The study highlighted that 18F-PSMA-1007 PET/CT exhibited a higher detection rate in patients with lower PSA levels, which directly influenced the initiation of systemic androgen deprivation therapy (ADT) and chemotherapy in patients with extensive metastatic disease. This emphasizes 18F-PSMA-1007 PET/CT’s role in identifying recurrence and guiding systemic treatment decisions, particularly in patients where locoregional salvage approaches may not be sufficient.

## 4. Discussion

This scoping review evaluated the current evidence on the diagnostic and prognostic capabilities of 18F-PSMA-1007 PET/CT for initial staging and detecting recurrent prostate cancer. A total of 35 articles were included in the review, predominantly from China and Europe, with fewer from other Asian countries, Australia, and Africa. Most were retrospective quantitative reviews with limited prospective studies, including one multicenter randomized control trial. Whilst 18F-PSMA-1007 PET/CT demonstrated high diagnostic accuracy and sensitivity for detecting locoregional and distant prostate cancer, its specificity, particularly for bone metastasis, is lower when compared to other imaging modalities. Its high accuracy for locoregional disease is attributed to favorable imaging properties, such as reduced urinary excretion. For patients with BCR, 18F-PSMA-1007 PET/CT is effective at low PSA levels, with detection efficacy increasing alongside PSA values. Compared to other imaging modalities in either setting, 18F-PSMA-1007 PET/CT generally offers improved performance but is limited by poor interreader variability and higher rates of equivocal lesions.

18F-PSMA-1007 PET/CT has demonstrated superior diagnostic accuracy compared to conventional imaging techniques for the initial staging of prostate cancer, specifically in detecting clinically significant lesions, seminal vesical invasion and pelvic lymph node metastases [19,22]. The ability to detect small oligometastatic lesions missed by conventional imaging has led to patient upstaging in several studies, thereby influencing clinical management and underscoring its importance in early detection and treatment planning [19]. This enhanced sensitivity is attributed to favorable imaging characteristics, particularly the lower urinary excretion of the PSMA-1007 tracer due to hepatobiliary clearance; however, incidental high urinary uptake has been reported in up to 32.4% of patients, limiting its consistency [50,51]. Additionally, poor specificity for bone lesions and the resultant interreader variability signify challenges to its routine use in primary staging [52]. Unlike 68Ga-PSMA-11 PET/CT, which is predominately excreted via the urinary tract, 18F-PSMA-1007 demonstrates lower renal clearance resulting in increased physiological bone uptake that may not correlate with metastatic disease. Several studies have cited false-positive bone lesions, particularly in patients with benign conditions such as degenerative changes, fractures and inflammatory processes [53,54,55]. This is significant when the clinician attempts to differentiate true osseous metastases from benign findings. To address these documented limitations, combining 18F-PSMA-1007 PET/CT with other imaging modalities, such as multiparametric MRI, has been proposed, demonstrating improved diagnostic accuracy, particularly for equivocal PIRADS 3 lesions, as demonstrated by Privé et al. (2024) [22,56]. Whilst these combined approaches are promising, larger prospective studies with greater sample sizes are needed to validate their efficacy and establish their role in refining the staging of prostate cancer prior to definitive therapy.

The diagnosis of biochemical recurrence (BCR) of Prostate cancer is a common diagnosis within 10 years of curative therapy, necessitating precise assessment and ongoing treatment planning [14]. 18F-PSMA-1007 PET/CT has demonstrated consistently high detection efficacy for PSA levels above 1.0ng/mL and moderate rates for below 1.0 ng/mL, outperforming other PSMA radiotracers in matched comparisons [33,41,44,46]. However, its diagnostic performance diminishes at lower PSA thresholds as highlighted by systematic reviews [57]. Despite this limitation, 18F-PSMA-1007 PET/CT can stratify risk and guide clinical decision-making at these thresholds, emphasizing its potentially nuanced role in BCR management [48]. Systematic reviews have also found 18F-PSMA-1007 PET/CT to have superior detection rates compared to commonly used radiotracers, though publication bias may have influenced these findings [58]. When compared to other imaging modalities, such as 18F-Choline PET/CT and 68Ga-PSMA-11 PET CT, 18F-PSMA-1007 PET/CT offers improved sensitivity; however, its limitations in detecting bone metastases and its tendency for uptake in the associated osseous benign lesion requires interpretation by experienced nuclear physicians to mitigate false positives [59]. These various PSMA radiotracers display distinct pharmacokinetics and biodistribution varying in their excretion methods. For instance, 18F PSMA labeled ligands generally offer a higher spatial resolution whilst 68Ga-PSMA-11 is more widely available due to generator-based production [60]. Therefore, the choice of tracer is guided by clinical context, imaging priorities and institutional availability. As such, the false positives of 18F-PSMA-1007 particularly in the context of bone lesions and non-malignant uptake, can offset the advantages of its superior sensitivity. Such limitations raise concerns about its suitability as a first-line imaging modality for detecting recurrent prostate cancer. Instead, 18F-PSMA-1007 PET/CT may find a more suitable role as a second-line imaging tool where there is difficulty resolving PSMA expression from urinary excretion as opposed to prostate cancer adjacent to urinary structures. This more target approach to the use of 18F-PSMA-1007 PET/CT could enhance its clinical impact whilst mitigating the risk of poorer specificity with skeletal lesions and interpretive challenges.

Furthermore, the integration of 18F-PSMA-1007 PET/CT regarding salvage therapy planning represents a paradigm shift in the management of biochemical recurrence of prostate cancer. Its high sensitivity for detecting recurrence at low PSA levels [45], ability to refine radiotherapy targeting [47], and influence on systemic therapy decisions [42] make it a valuable imaging modality in post-treatment disease monitoring. Studies have demonstrated its superiority over conventional imaging techniques such as 18F-Choline PET/CT, highlighting its role in reducing diagnostic uncertainty and enhancing the precision of salvage treatment strategies [44]. Despite this, more prospective studies are required to further establish its position as a standard imaging tool in guiding personalized salvage therapy approaches for prostate cancer recurrence.

Several limitations are present in this Scoping review. Many of these included studies were retrospective, which may introduce potential selection bias and hence restrict the generalizability of the findings. Sample sizes vary, however in many instances such as Lengana et al. (2021) and Zhou et al. (2022) were small, limiting the statistical power to detect meaningful differences between study subgroups or imaging modalities. Additionally, many of the studies differed in terms of tracer dosages, interpretation criteria, and differences in imaging protocols, thereby making it difficult to provide more in-depth direct comparisons. These limitations in the current literature surrounding 18F-PSMA-1007 PET/CT outline the need for multicenter, prospective and randomized trials across many aspects of its use to further comment on future clinical use and viability.

## 5. Conclusions

18F-PSMA-1007 PET/CT is a valuable tool in the arsenal of the clinical urological oncologist for the staging and detection of primary and recurrent prostate cancer. Current evidence highlights its superior diagnostic accuracy and sensitivity for primary staging, demonstrating a reasonable ability to upstage cancer at initial staging by detecting small oligometastatic disease. When combined with other imaging modalities, particularly mpMRI, 18F-PSMA-1007 PET/CT is superior in downgrading PIRADS 3 equivocal lesions. Moreover, 18F-PSMA-1007 PET/CT demonstrates enhanced detection efficacy for recurrent disease at lower PSA levels, compared to conventional imaging modalities and other commonly used PSMA radiotracers, and hence can help guide salvage therapies. However, notable limitations include its poor specificity for bone lesions, and inconsistent urinary excretion patterns, with additional prospective multicenter trials required to clearly establish its role in prostate cancer staging.

## Figures and Tables

**Figure 1 cancers-17-01049-f001:**
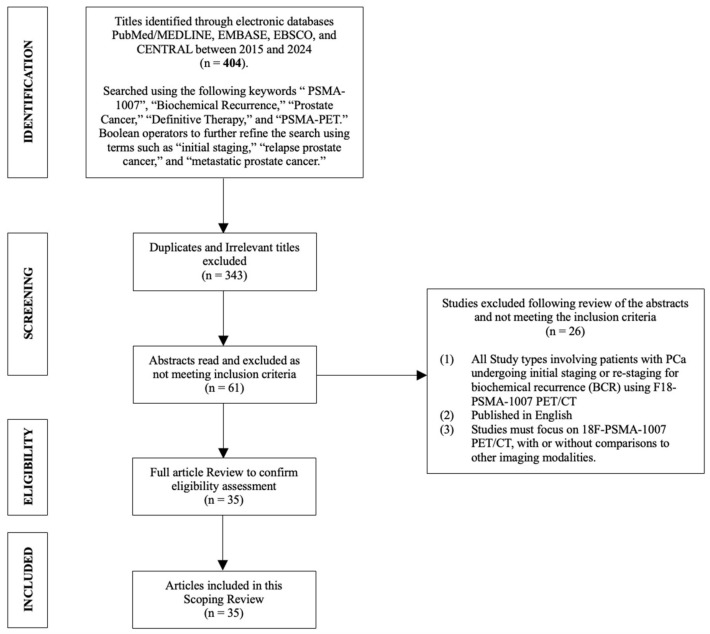
Flow diagram including search strategy.

**Table 1 cancers-17-01049-t001:** Scoping review articles summary for initial staging of PCa.

Study	Aims/Objectives	Design	Sample Characteristics	Key Findings
1	Evaluate 18F-PSMA-1007 PET/CT as a biomarker for pathological upgrading.	Retrospective	152 patients diagnosed with PCa, pre-RP imaging	Pathological upgrading in 26.97% of cases; prostate volume (PV) and PSMA uptake were independent risk factors.
2	Assess the diagnostic value of 18F-PSMA-1007 PET/CT with PSA-derived indicators for gray zone prostate cancer.	Retrospective	107 patients with PSA 4–10 ug/L.	PSA Density (PSAD) and SUVmax were significantly higher in PCa cases, aiding the differentiation of malignant vs benign pelvic lymph nodes.
3	Investigate 18F-PSMA-1007 PET/CT in equivocal PIRADS 3 lesions.	Prospective	75 participants in three PIRADS groups: Group1: PIRADS 1 or 2 Group2: PIRADS 3 Group3: PIRADS4 and 5	18F-PSMA-1007 PET/CT correctly classified 17/26 PIRADS 3 lesions (NPV 93%). Detected 1 additional csPCa lesion missed by mpMRI.
4	Compare diagnostic efficacy of 18F PSMA-1007 PET/CT and mpMRI for pelvic LN metastasis.	Retrospective	30 patients. Median age = 68.4 years post-RP with ePLND.	18F-PSMA-1007 PET/CT had higher sensitivity and NPV for LN detection, while mpMRI had greater specificity.
5	Compare 18F PSMA 1007 PET/CT Vs 68 Ga PSMA 11 PET/CT for initial staging.	Prospective	40 high-risk PCa patients	18F-PSMA-1007 PET/CT detected 3 additional regional LNs and 7 skeletal lesions, leading to upstaging in 2/5 patients.
6	Evaluate diagnostic accuracy of 18F-PSMA-1007 PET/CT for PCa metastases.	Retrospective	257 patients. Median tPSA 16.34, median Gleason score 8.	Diagnostic accuracy correlated with GS, PSA levels and D’Amico Risk classification.
7	Compare interreader agreement between 18F-DCFPyL and 18F-PSMA-1007 PET/CT.	Prospective	240 matched patients.	18F-PSMA-1007 PET/CT had higher detection rates for local recurrence but lower agreement for skeletal lesions.
8	Compare 18F-PSMA-1007-PET vs mpMRI and histology for primary staging of PCa.	Retrospective	53 patients, 66 suspicious mpMRI-identified lesions.	18F-PSMA-1007 PET/CT staged SV invasion better than mpMRI (90% vs. 76%); mpMRI was superior for ECE (90% vs. 57%).
9	Compare 18F-PSMA-1007 PET/CT with conventional imaging	Prospective	79 men, 5 imaging exams per patient (BS, CT, 99mTc-HMDP SPECT CT, WBMRI, 18F-PSMA-1007 PET/CT).	18F-PSMA-1007 PET/CT influenced treatment in 18% of cases. Detected 11/20 metastases missed by conventional imaging.
10	Investigate the role of 18F-PSMA-1007 PET/CT in diagnosing primary/metastatic PCa.	Retrospective	18 newly diagnosed PCa patients.	18F PSMA-1007 PET/CT altered treatment in 47.1% of cases. Detected metastases in 72.2%.
11	Compare 18F-PSMA-1007 PET/CT and pelvic MRI for PCa detection.	Retrospective	41 patients, median PSA 136.1.	PET/CT had higher sensitivity (95.1%) and accuracy (95.1%) vs. MRI (82.9%).
12	Evaluate 18F-PSMA-1007 PET/CT vs. mpMRI for local staging.	Retrospective	10 patients	18F-PSMA-1007 PET/CT had near-total agreement with RP histology (93%) vs. mpMRI (87%).
13	Evaluate the diagnostic performance of 18F-PSMA-1007 PET/CT for intraprostatic PCa	Retrospective	Group A—117 patients. Confirmed PCa on biopsy, Mean age 69 ± 6.9. Group B—76 participants. Prospective cohort. Used to validate results. Mean age 71 ± 8.2.	18F-PSMA-1007 PET/CT prostate cancer detection rate = 96.6% (113/117). Sensitivity = 91.2%, PPV = 89.8% for intraprostatic lesions.
14	Compare interobserver variability of 3 PSMA radiotracers: 68Ga-PSMA-11, 18F-PSMA-1007, and 18F-DCFPyL, in primary prostate cancer (PCa) staging	Retrospective	584 patients, three radiotracer groups.	18F-PSMA-1007 had higher variability for bone metastases than other tracers due to non-specific uptake.
15	Compare 18F-PSMA-1007 PET/CT vs. 99Tcm-MDP SPECT/CT for bone metastases.	Retrospective	77 patients (65 initial stagings, 12 restagings).	PET/CT had higher sensitivity (100%) and accuracy (98.7%) compared to SPECT/CT (76.6%).
16	Compare 18F-PSMA-1007 and 68Ga-PSMA-11 for TNM staging.	Prospective	50 patients, mixed staging/restaging.	PET/CT had higher uptake in local recurrence, nodal, and distant lesions, with lower bladder activity.

1—Zheng et al., (2023) [15]; 2—Guo et al., (2023) [16]; 3—Privé et al., (2024) [17]; 4—Liu et al., (2022) [18]; 5—Chandekar et al., (2023) [19]; 6—Bai et al., (2022) [20]; 7—Wondergem et al., (2021) [21]; 8—Privé et al., (2021) [22]; 9—Anttinen et al., (2021) [23]; 10—Li et al., (2020) [24]; 11—Ye et al., (2024) [25]; 12—Kesch et al., (2017) [26]; 13—Luo et al., (2023) [27]; 14—Hagens et al., (2022) [28]; 15—Chen et al., (2023) [29]; 16—Pattison et al., (2022) [30].

**Table 2 cancers-17-01049-t002:** Scoping review articles summary for detection of recurrent PCa.

Study	Aims/Objectives	Design	Sample Characteristics	Key Findings
17	Investigate 18F-PSMA-1007 PET/CT in detecting BCR in PCa patients.	Retrospective	71 patients following RP. Median age = 67. Mean PSA = 1.27	18F-PSMA-1007 PET/CT detected recurrence in 79% of cases, with higher detection rates in higher Gleason scores. 50% of patients with PSA < 0.5 had a positive scan.
18	Compare 18F-PSMA-1007 PET/CT vs. 68Ga-PSMA-11 PET/CT in BCR Detection.	Prospective	21 patients with BCR following definitive therapy	18F-PSMA-1007 PET/CT had superior sensitivity (88.9% vs. 44.4%), and specificity (100% vs. 83.3%) than 68Ga-PSMA-11 PET/CT.
19	Compare 18F-PSMA-1007 vs. 18F-Choline PET/CT in BCR.	Prospective, Randomized Control Trial	186 BCR patients post-primary treatment.	18F-PSMA-1007 had a higher detection rate (84%) than 18F-Choline (69%), especially for low PSA levels. Superior for detecting metastatic lesions.
20	Evaluate 18F-PSMA-1007 PET/CT vs. CI in detecting nonregional LN metastases.	Retrospective	224 with mHSPC patients	18F-PSMA-1007 PET/CT and CI had 61.6% concordance. 37/94 cases were upstaged by PET/CT, improving treatment stratification.
21	Assess diagnostic accuracy of 18F-PSMA-1007 PET/CT for PCa recurrence.	Retrospective	177 post-treatment patients were included,	Overall sensitivity: 95%, specificity: 89%. PPV: 97% for local recurrence, 93% for pelvic LNs, but lower for bone metastases (79%).
22	Evaluate 18F-PSMA-1007 PET/CT in BCR detection at low PSA.	Prospective	45 patients, PSA <2.0 ng/mL	Sensitivity: 100%, specificity: 92.8%, accuracy: 97.8%. Detection rate was 50% for PSA <0.5 ng/mL and 88.2% for PSA 1–2 ng/mL.
23	Assess 18F-PSMA-1007 PET/CT in BCR not detected with standard imaging.	Prospective	28 BCR patients	Detection rates: 66.7% (PSA 0.1–0.5 ng/mL), 85.7% (PSA 0.5–1.0 ng/mL), 100% (PSA > 1.0 ng/mL). PET identified recurrence in 26/28 cases.
24	Investigate false positives in 18F-PSMA-1007 PET/CT.	Prospective	102 patients	18F-PSMA-1007 PET/CT detected more benign lesions than 68Ga-PSMA-11, but also identified 124 recurrent PCa lesions.
25	Assess scan positivity predictors for 18F-PSMA-1007 PET/CT.	Retrospective	137 BCR patients	Scan positivity: 80%. PSA level and PSAV were positive predictors. Prior ADT correlated with more bone and LN involvement.
26	Compare 18F-PSMA-1007 vs. 68Ga-PSMA-11 for cost-effectiveness and clinical outcomes.	Retrospective	240 recurrent PCa patients	PET positivity: 91.8% (68Ga) vs. 86.9% (18F). 18F had more equivocal findings (17.2% vs. 8.25%). 68Ga had a higher PPV (0.99 vs. 0.86).
27	Evaluate 18F-PSMA-1007 PET/CT diagnostic role in BCR.	Prospective	12 patients, median PSA 0.60 ng/mL	75% had PET-positive lesions. SUVmax significantly increased from 1 to 3 h post-injection (7.00 vs. 11.34, *p* < 0.001).
28	Evaluate impact of 18F-PSMA-1007 PET/CT on systemic therapy decisions.	Retrospective	264 post-radical treatment patients Mean PSA 1.6 ng/mL	18F PSMA 1007 PET, Positivity rate = 87.5% (112/128). Mean PSA 7.04 ± 18.56 ng/mL Pre-scan PSAs were significantly different for 18F PET positive vs 18F PET negative scan. Mean and median PSAs were significantly higher in positive scans as opposed to negative scans.
29	Evaluate 18F-PSMA-1007 PET/CT performance in early BCR.	Prospective	40 post-treatment PCa patients	Detection rate: 60% overall. 39% for PSA < 0.5 ng/mL, 55% for PSA 0.5–1.0 ng/mL, and 100% for PSA ≤ 2.0 ng/mL. Sensitivity: 100%, specificity: 94.4%.
30	Evaluate the diagnostic performance of 17F Fluorocholine (FCH) vs 18F PSMA 1007 PET/CT in BCR at low PSA levels.	Quantitative, Prospective	80 patients	18F PSMA-1007 PET/CT Positive results = 60% Equivocal Lesions = 27.5% Negative results = 12.5% 18F FCH Scans Positive results = 5% Equivocal results = 37.5% Negative results = 57.5% 18F PSMA 1007 upgraded 18F FCH results in 70% of cases and detected more lesions (184 vs. 63, *p* = 0.0006). Lesion types: 1. 18F PSMA 1007 PET/CT ◦ Local relapse = 9% ◦ Lymph Node = 58% ◦ Bone = 33% 2. 18F FCH PET/CT ◦ Local relapse = 5% ◦ Lymph Node = 89% ◦ Bone = 6% Highly suspicious lesions 18F PSMA 1007 = 74% 18F FCH = 11% SUVmax values in 18F PSMA 1007 were significantly higher in suggestive lesions than for equivocal lesions (median SUVmax: 3.6 vs. 2.5, *p* < 0.00001).
31	Assess 18F-PSMA-1007 PET/CT for early BCR.	Retrospective	46 BCR patients, median PSA 1.6 ng/mL	Detection rate: 52.2%. Oligometastatic disease in 32.5% (mostly lymph nodes). PSA cutoff for optimal detection: 1.3 ng/mL.
32	Evaluate 18F-PSMA-1007 PET/CT in post-RP BCR.	Retrospective	251 BCR patients	Detection rate: 94% (PSA ≥ 2 ng/mL), 74.5% (PSA 0.5–1 ng/mL). PET/CT identified lymph node and bone metastases in high Gleason scores.
33	Investigate PET/CT positivity in high-risk vs. low-risk BCR patients.	Retrospective	71 included patient Median PSA 1.43 ng/mL	Higher detection in high-risk (72.4%) vs. low-risk (38.0%) patients. Similar proportion of pelvic-confined disease in both groups (24.1% vs. 23.1%, *p* = 0.935).
34	Evaluate the effectiveness of 18F PSMA-1007 PET/MRI in PSA levels < 0.5 ng/mL following localized treatment.	Prospective	35 patients with PCa with a PSA < 0.5 ng/mL.	25/35 (71.4%) had positive PET/MRI results. 15/35 (42.9%) had local recurrence PET/MRI findings guided treatment decisions in 80% of patients
35	Evaluate PET/CT accuracy for lymph node staging.	Retrospective	96 patients. Mean age = 69.5 years Mean time between PET/CT and surgery = 47.65 ± 35.87 days	Positive LNs were identified in 34.4% of patients. Diagnostic performance of 18F PSMA-1007 PET/CT: Sensitivity = 81.7% Sepcificity = 99.6% PPV = 92.4% NPV = 98.9% Performance metrics were specific to LNs > 3 mm.

17—Zhou et al., (2022) [31]; 18—Lengana et al., (2021) [32]; 19—Panagiotidis et al., (2023) [33]; 20—Jiang et al., (2023) [34]; 21—Mingels et al., (2022) [35]; 22—Tian et al., (2020) [36]; 23—Watabe et al., (2021) [37]; 24—Rauscher et al., (2020) [38]; 25—Ahmadi Bidakvidi et al., (2021) [39]; 26—Alberts et al., (2022) [40]; 27—Giesel et al., (2018) [41]; 28—Hoffmann et al., (2022) [42]; 29—Wikowska-Patena et al., (2020) [43]; 30—Witkowska-Patena et al., (2019) [44]; 31—Lengana et al., (2022) [45]; 32—Giesel et al., (2019) [46]; 33—Poterszman et al., (2024) [47]; 34—Garcia et al., (2024) [48]; 35—Sprute et al., (2024) [49].

## Data Availability

The original contributions presented in this study are included in the article. Further inquiries can be directed to the corresponding author.

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
