# Peer review of "The Role of 18F PSMA-1007 PET/CT in the Staging and Detection of Recurrence of Prostate Cancer, A Scoping Review"

_cancers, 2025, doi:10.3390/cancers17061049_

Round 1
Reviewer 1 Report
Comments and Suggestions for Authors
A very well-known topic without any significant aid to the relevant literature..
A review, kind of meta-analysis, that focused on prior paper results and based on their yields without any novelty, originality.
No figures, no tables, poor presentation.
Reviewer 2 Report
Comments and Suggestions for Authors
I suggest as minor revision to improve the table 1 by shortening it.
Reviewer 3 Report
Comments and Suggestions for Authors
The present manuscript is a well written and interesting work reporting a complete review of the current available litarature on the capability of 18F-PSMA-1007 PET/CT in the settings of initial staging and detection of recurrent disease for patients with prostate cancer.
The analysis is exhaustibe and the reported contents are complete and useful in the clinical scenario.
As a general comment, being the information reported so numerous and detailed, I would suggest, to revise the paper with a greater effort of synthesis and schematization, to present the data in a more clear way to the reader. In particular, both Staging and Re-staging papers are included. Both in the abstract and in the main text, a clear differentiation between the two different chapters should be provided. It has been mentioned also the role of PET/CT to guide salvage therapy. Evaluate to estrapolate the papers dealing with that aim, to define a specific focused chapter.
In addition, in any single chapter (Staging and Re-staging), a further schematisation should be proposed, both in the Method and in the corresponding Results session.
As an example:
In Staging: clearly separate data on PET/CT Accuracy based on histological validation, on PET/CT Accuracy based on histological validation; the site of neoplastic disease (primary cancer, LN, bone), the comparison with other Imaging modalities (MR)….
In Restaging: clearly report data as based on the different instrumental modalities used as gold standard to evaluate accuracy, the different evaluations as based on the different values of PSA… etc…
The two Tables reporting all the papers included (Staging and Re-Staging) are very informative. However, they are very difficult to be read and in particular to provide a take-home-message. I would propose to revise the whole Tables reporting only the very significant messages, in a very brief and synthetic way (approximately one row for each paper). In addition, the number of papers included in each table should be reported in the legend title, and listed with consecutive numbers.
Reviewer 4 Report
Comments and Suggestions for Authors
Dear Authors,
this is an interesting paper that focuses on a wide field of current researches. Some issues are however presents:
- in Tables 1 and 2 information about the dose of radiopharmaceuticals injected is missing. In addition, it is not clear the meaning of "quantitative" in the design column, since PET/CT has only semiquantitative parameters;
- an evaluation of the quality of the papers included in the review should be performed, for example by using the QUADAS tool;
- adding some subheadings to the different parts of the Results section would be useful;
- a metanalysis for sensitivity and specificity could be useful to strenghten your findings;
- even though in the discussion the problem related to the false negative findings in the bone has been cited, unspecific bone uptakes (UBU) have not been discussed and should therefore be included;
- line 323-327 "When compared to 323 other imaging modalities, such as 18F-Choline PET/CT of 68Ga-PSMA-11 PET CT, 18F-324 PSMA-1007 PET/CT offers improved sensitivity; however, its limitations in detecting 325 bone metastases and its tendence for uptake in benign lesions interpretation by experi-326 enced nuclear physicians to mitigate false positives [57]": the difference between different PSMA forms should be described and a brief comparison should be discussed.
